# Rates of Intracranial Hemorrhage in Mild Head Trauma Patients Presenting to Emergency Department and Their Management: A Comparison of Direct Oral Anticoagulant Drugs with Vitamin K Antagonists

**DOI:** 10.3390/medicina56060308

**Published:** 2020-06-23

**Authors:** Gabriele Savioli, Iride Francesca Ceresa, Sabino Luzzi, Cristian Gragnaniello, Alice Giotta Lucifero, Mattia Del Maestro, Stefano Marasco, Federica Manzoni, Luca Ciceri, Elia Gelfi, Giovanni Ricevuti, Maria Antonietta Bressan

**Affiliations:** 1Emergency Department, Fondazione IRCCS Policlinico San Matteo, 27100 Pavia, Italy; irideceresa@gmail.com (I.F.C.); luca.ciceri02@universitadipavia.it (L.C.); gelfielia@gmail.com (E.G.); mita.bressan@gmail.com (M.A.B.); 2Department of Clinical-Surgical, PhD School in Experimental Medicine, Diagnostic and Pediatric Sciences, University of Pavia, 27100 Pavia, Italy; m.delmaestro@smatteo.pv.it; 3Neurosurgery Unit, Department of Clinical-Surgical, Diagnostic and Pediatric Sciences, University of Pavia, 27100 Pavia, Italy; sabino.luzzi@unipv.it (S.L.); alicelucifero@gmail.com (A.G.L.); stefanom92@outlook.it (S.M.); 4Neurosurgery Unit, Department of Surgical Sciences, Fondazione IRCCS Policlinico San Matteo, 27100 Pavia, Italy; 5Department of Neurological Surgery, University of Illinois at Chicago, Chicago, IL 60656, USA; cristian.neuro@gmail.com; 6Clinical Epidemiology and Biometry Unit, Fondazione IRCCS Policlinico San Matteo, 27100 Pavia, Italy; f.manzoni@smatteo.pv.it; 7Department of Drug Science, University of Pavia, Italy, -Saint Camillus International University of Health Sciences-Rome-Italy, 27100 Pavia, Italy; giovanni.ricevuti@unipv.it

**Keywords:** mild head trauma, anticoagulated patients, emergency department, direct anticoagulant drugs, DOAC, NOAC, new anticoagulant drugs, vitamin K antagonist anticoagulants, VKAs, mild head injury, minor head trauma, minor head injury, intracranial hemorrhage, surgery, ED revisit rates, intrahospital mortality, computed tomography, risk management, hemorrhage, observation

## Abstract

*Background and objectives*: Anticoagulants are thought to increase the risks of traumatic intracranial injury and poor clinical outcomes after blunt head trauma. The safety of using direct oral anticoagulants (DOACs) compared to vitamin K antagonists (VKAs) after intracranial hemorrhage (ICH) is unclear. This study aims to compare the incidence of post-traumatic ICH following mild head injury (MHI) and to assess the need for surgery, mortality rates, emergency department (ED) revisit rates, and the volume of ICH. *Materials and Methods*: This is a retrospective, single-center observational study on all patients admitted to our emergency department for mild head trauma from 1 January 2016, to 31 December 2018. We enrolled 234 anticoagulated patients, of which 156 were on VKAs and 78 on DOACs. Patients underwent computed tomography (CT) scans on arrival (T0) and after 24 h (T24). The control group consisted of patients not taking anticoagulants, had no clotting disorders, and who reported an MHI in the same period. About 54% in the control group had CTs performed. *Results*: The anticoagulated groups were comparable in baseline parameters. Patients on VKA developed ICH more frequently than patients on DOACs and the control group at 17%, 5.13%, and 7.5%, respectively. No significant difference between the two groups was noted in terms of surgery, intrahospital mortality rates, ED revisit rates, and the volume of ICH. *Conclusions*: Patients with mild head trauma on DOAC therapy had a similar prevalence of ICH to that of the control group. Meanwhile, patients on VKA therapy had about twice the ICH prevalence than that on the control group or patients on DOAC, which remained after correcting for age. No significant difference in the need for surgery was determined; however, this result must take into account the very small number of patients needing surgery.

## 1. Introduction

Minor head injury (MHI) is one of the most common emergency room cases in Italy and abroad, with about 75% presenting with head trauma. Minor head injury is defined as a patient with a history of loss of consciousness, amnesia, or disorientation and a Glasgow Coma Scale (GCS) score of 13–15 [1,2]. Anticoagulant medications are commonly used for a variety of indications [3,4,5,6,7,8,9], and it is widely believed that preinjury use of these medications increases the risk of traumatic intracranial injury and worsens clinical outcomes after blunt head trauma [10,11,12,13]. This belief is based largely on biological plausibility and retrospective cohort studies [14,15,16,17]. There has been increasing interest in this topic, as the use of direct oral anticoagulant (DOAC) therapy is becoming more widespread. Prospective studies [18,19,20,21,22,23,24] and literature reviews [25,26,27] have demonstrated a higher safety profile for intracranial hemorrhage (ICH) following an MHI in patients treated with DOAC rather than those treated with vitamin K antagonists (VKAs). The advantage of this new class of anticoagulants in elderly patients [28,29] has also been determined.

Direct oral anticoagulants (DOACs) have partially replaced vitamin K antagonists (VKAs) in the prevention of thromboembolism. They provide a similar prophylactic effect against stroke or peripheral arterial and venous embolism, but with a significantly lower incidence of spontaneous intracerebral hemorrhage [27,30,31,32,33]. Moreover, DOACs do not require the monitoring of anticoagulation activity, and they have a shorter half-life than VKA. Despite these advantages, physicians are reluctant to prescribe DOAC, since their antidotes are not broadly available [34].

Recent studies, however, have shown conflicting data for patients on DOACs; this is in terms of higher risk for ICH, ICH progression, or death [23,24] and regarding hematoma expansion, mortality, and rates of operational intervention in patients on DOACs compared to VKAs [35,36,37]. Some recent studies have also caused doubts about the belief that anticoagulant use is predictive of traumatic intracranial hemorrhage (ICH) after adjusted analysis, especially by age [38,39].

This study aimed to assess the different bleeding risk profiles of patients taking direct oral anticoagulants and traditional vitamin K antagonists who are admitted to the Emergency Department for MHI. The primary goal was to determine the difference in the incidence of recurrent or worsened post-traumatic ICH following MHI. The secondary objective was to assess the need for surgery. We also examined the intrahospital mortality rates, emergency department (ED) revisit rates, and the volume of ICH.

## 2. Materials and Methods

### 2.1. Study Design

We conducted a retrospective and monocentric observational study of all patients admitted to the emergency department of the IRCCS Polyclinic San Matteo Foundation in Pavia, Italy, from 1 January 2016, to 31 December 2018, for mild head trauma. The primary endpoint was the diagnosis of recurrent or worsened post-traumatic ICH during observation in the emergency department. The secondary objective was to assess the need for surgery. We also examined the intrahospital mortality rates, ED revisits rates, and the volume of ICH. The morphological assessment of ICH was also performed by comparing brain computed tomography (CT) (with critical volume assessment, Marshall scale, location, and size) undertaken by a single experienced neuroradiologist.

### 2.2. Inclusion and Exclusion Criteria

All the patients with the following characteristics were included in the study: (1) with mild head trauma, (2) who were on anticoagulation therapy, (3) age 18 or above, and (4) have a Glasgow Coma Scale (GCS) score of >13 on arrival in ED. We also included a control group consisting of patients with mild head trauma, age > 18, and GCS score of >13 on arrival at the ED without any anticoagulants, anti-aggregation therapy or others clotting disorders. We excluded patients on antiplatelet or heparin therapy and who did have clotting disorders from both the VkAs and DOACs group. For the DOACs group we recruited patients in therapy with apixaban, dabigatran, edoxaban and rivaroxaban. Regarding the VkAs group, we recruited patients in therapy with acenocumarol and warfarin.

The exclusion criteria were as follows: ICH with no history of head trauma, and GCS (GCS) score of <13 on arrival in ED.

### 2.3. Study Population

Eligible patients were identified in the electronic database through discharge diagnosis codes corresponding to “cranial trauma”, “ICH”, and “skull/face/neck trauma.” 

For every patient, we collected their demographic data (gender and age), dynamics of trauma, blood pressure, heart rate, oxygen saturation, GCS, signs and symptoms, hematochemical examinations (in particular hemochrome and clotting examinations), length of stay (LOS) in the ED, development of ICH, need for hospitalization or surgery, intrahospital mortality related to head trauma, and ICH volume. All the medical records have been accurately viewed and evaluated, and all CTs were thoroughly reviewed. For patients on anticoagulant therapy with a positive CT scan, we determined the topography of the hemorrhage (intraparenchymal, subdural, epidural, or subarachnoid), its size, and its gravity according to the Marshall scale. All the collected data were stored on a spreadsheet using the Microsoft Excel program and were later used for statistical analysis. 

We included 236 consecutive patients in this study who reported MHI, of which 156 were on VKA and 78 on DOAC. The control group was composed of 1443 patients who were not on any therapy.

### 2.4. Management of Patients with Mild Head Trauma

#### 2.4.1. ED Organization

Our ED is a major trauma reference center. We are the only hospital in the province with neurosurgery services. Our neurosurgery service is 24 h.

#### 2.4.2. Criteria for Computed Tomography (CT); Observation; Neurosurgical Visit; Patient’s Suitability for Discharge

Patients on DOAC or VKA therapy were observed within 24 h. Patients with higher dynamic criteria were observed for 12–24 h based on clinical judgment. All the other patients were observed for 6 h. The clinical criteria for performing CTs during the observation period, which are in accordance with our company protocol and both Canadian [1,40] and Italian guidelines [41], are presented in Table 1. The dynamic criteria for major trauma are listed in Table 2. 

All patients on DOAC or VKA therapy underwent a CT on arrival and after the 24-h observation period (T0 and T24, respectively). 

At the end of the observation period, in the absence of symptoms and with negative CTs (when performed), patients are discharged home accompanied by their caregiver and provided with a module of instructions for home observation appropriate for mild head trauma. In the module, it recommends that they return to our ED for a CT scan if the listed symptoms occur (Table 1). All patients who developed ICH had neurosurgical counseling before hospitalization. Patients on anticoagulant therapy who had ICH were given with the antidote (vitamin K for patients on VKAs and idarucizumab for patients on dabigatran) and prothrombin complex concentrate (PCC). 

### 2.5. Statistical Analysis

The analyses were carried out using the appropriate univariate and multivariate models of logistic regression (with age correction) to test the association between therapy and the ICH rate. Continuous variables were described with mean and standard deviation, while the qualitative variables were expressed with counts and percentages. The results were provided as OR with its 95% confidence interval. Comparisons between groups for continuous variables were made using the Student’s *t*-test or non-parametric Mann–Whitney U test. Associations between qualitative variables were examined using Fisher’s exact test. The significance level was set at alpha = 0.05 (statistical significance for *p*-value < 0.05), and all tests were two-tailed. The analyses were conducted using the Stata software, version 14 (Stata Corporation, College Station, TX, USA, 2015).

## 3. Results

### 3.1. Study Population

The two populations on anticoagulant therapy (DOAC and VKAs) had no statistically significant differences in terms of age, gender, GCS, blood pressure, heart rate, and the prevalence of the major dynamics of trauma. The control group was younger and had a higher rate of major trauma dynamics (*p* < 0.01) (Table 3).

The two populations on anticoagulant therapy stayed longest in the ED. The percentage of patients who underwent CT is shown in Table 3a.

Mechanisms of injury are summarized in Table 3b.

#### 3.1.1. Patients on Direct Oral Anticoagulants (DOACs)

We enrolled 78 patients on DOAC who reported with MHI. In patients receiving direct anticoagulants, 36 (46.75%) were taking apixaban, 25 (32.47%) rivaroxaban, 12 (15.58%) dabigatran, and 4 (5.19%) edoxaban. 

#### 3.1.2. Patients on Traditional Vitamin K Antagonists (VKA)

Patients receiving dicoumarol treatment also had their international normalized ratio for prothrombine time (INR) values measured. Out of the 156 patients, INR was available in 144 (92%) cases, with a total average value of 2.53 ± 1.15. Only 16 patients (11%) were below the therapeutic range, while 102 (71%) had an INR between 1.5 and 3 and for 26 patients (18%) it was above 3. 

### 3.2. Hemorrhagic Complications

All positive CT images of patients on oral anticoagulant therapy were individually reanalyzed. From these, data were obtained in terms of the type of hemorrhage, its size, and the Marshall classification.

#### 3.2.1. Patients on DOACs

The bleeding rate in this subset of patients was recorded to be at 5.13% (3.85% at T0; 1.28% at T24) (Table 6). Two ICH patients were on Apixaban, 1 on rivaroxaban; and 1 on dabigatran. There was, therefore, no statistically significant difference as regards ICH in DOAC patients compared to the control group (OR = 0.663; *p* > 0.05) (Table 4). No statistical difference was also determined even with age-corrected regression compared to controls (OR = 0.438; *p* > 0.05) (Table 5). Among patients on DOAC therapy, there were two cases of subdural hemorrhage (50%), one case of subarachnoid hemorrhage (25%) and another of epidural hemorrhage (25%). The severity of the bleeding was evaluated using the Marshall scale. In DOAC patients, there was an average score of 3 (SD = 2) and a median score of 2, while the average volume of bleeding was 10.4 cm^3^ (SD = 9.9). 

#### 3.2.2. Patients on VKAs

The bleeding rate in this subset consisting of 27 patients was recorded to be at 17.3% (14.10% at T0; 3.20% at T24) (Table 6). The risk of these patients was about twice as high as for patients who did not take any therapy, reaching statistical significance (OR = 2.57; *p =* 0.000) (Table 4). Age-corrected regression remained statistically significant compared to patients not on therapy (OR = 1.655; *p* = 0.04) (Table 5). There were 10 (37.03%) cases of subarachnoid hemorrhage, 12 (44.44%) of subdural hemorrhage, and 7 (25.92%) of intraparenchymal hemorrhage. Of these, three were in mixed forms. On the Marshall scale, VKA patients had an average score of 3.04 (SD = 1.68), with a median value of 2. The average volume of bleeding in VKA patients was 15.8 cm^3^ (SD = 32.4). There was no statistically significant difference compared to DOAC; *p* > 0.05.

#### 3.2.3. Control Group: Patients Not on Therapy

Patients who were not on any therapy and had no clotting disorders developed ICH in 6.51% of cases. Of these, 6.44% were at T0, while 0.07% were positive at T24 (Table 6). Although they should not have had a second CT according to the protocol, they required a second CT based on the appearance of symptoms during the observation period. These were patients with greater trauma dynamics.

### 3.3. Need for Surgery

No patient on DOAC needed an operation, and only one patient on the VKA did. There was no statistically significant difference in surgery compared to patients in DOAC (*p* > 0.05) (Table 6). Only five patients in the control group needed surgery (Table 6).

### 3.4. Prolonged Observation and ED Revisit within 30 Days

The percentage of patients who exceeded the expected observation period was 15% in the control group (patients without any therapy), 20% in the VkAs group and 28% in the DOACs group. Of the patients on VKAs, only 10.9% revisited the ED, with 5.13% for new trauma and 5.77% for other non-traumatic problems (Table 6). None of these patients revisited the ED for reasons related to head trauma.

Of the patients on DOAC, 14.10% revisited the ED within the next 30 days, with 6.42% returning as a result of new trauma and 6.41% for other non-traumatic issues (Table 6). Only about 1.3% revisited the ED for reasons related to trauma, a mild wound complication, and the onset of a headache. All new CTs performed were negative.

In the control group, 7.13% revisited the ED within the next 30 days. There were 2.57% who suffered new trauma, and 3.32% had non-traumatic problems (Table 6). About 1.25% made an ED revisit for trauma-related reasons. These were mostly for minor wound complications and the removal of stitches. The few patients who returned due to symptoms (e.g., headache) had negative CT scans.

### 3.5. Intrahospital Mortality

None of the patients in this study died in the hospital (Table 6).

## 4. Discussion

### 4.1. Hemorrhagic Complications

It is important to note that our study population is made up of patients who, in real life, are admitted to ED for mild head trauma. Moderate and severe head traumas are excluded from our analysis. 

Regarding patients on dicoumarol treatment, we recorded a significantly higher bleeding rate (17.3%) compared to the other two groups. This trend tended to increase in the population that had an INR of more than three, where intracranial bleeding reached a prevalence of 27%. The increase in the ICH rate as the INR increases is in line with the findings of previous reports [42,43,44,45]. However, the figure has not been confirmed by all studies [46]. It should be noted that 18% of our population had subtherapeutic INR values; according to previous studies, a value of 1.5 is considered subtherapeutic [47]. We compared this cohort of patients with those not on any therapy using multivariate logistic regression, and the risk was increased by about 55%. 

Our result of the highest ICH figure being in VKA patients agrees with many studies in the literature [9,10,17,33,40,48,49,50,51,52,53,54,55,56].

This would be expected when anticoagulant therapy increases the hemorrhagic risk in itself, although head trauma can also promote trauma-induced coagulopathy and, consequently, ICH [35,57,58,59,60,61].

Because the population on VKAs was significantly older than the control group (*p* < 0.005), we performed a logistic regression, taking age into account. There was a significant increase (an average of 65.5%) in the risk of reporting bleeding in patients on VKA therapy compared to the control group (aOR-1.65, *p* = 0.048). In our population, patients on VKAs, therefore, had a higher prevalence of ICH even when correcting for age. In contrast, other studies have found it to be attenuated [38,39].

Patients on DOAC therapy developed ICH in only 5.1% of cases, which was not statistically significantly different from the control group. This data must be contextualized. The population on DOACs is likely to be composed of patients who are highly sensitized and who are likely to access the ED for relatively low levels of trauma, which healthy patients not on therapy would probably not do. No statistically significant difference was found between the control group and the DOAC group in the development of post-traumatic ICH (*p* = 0.122). It should also be noted that all patients in the DOAC group underwent CTs compared to only 54.7% of patients in the control group. 

Based on our data, DOACs have a better safety profile than VKAs in terms of ICH following MHI.

### 4.2. Patient Management

Patient management for those on VKAs and DOAC does not differ, and the observation period is 24 h. This involves resource utilization and a longer stay in ED, as demonstrated by the long LOS (Table 3). However, the need for a control head CT scan in patients on anticoagulation therapy that do not display neurological deterioration is disputable, since the incidence of late ICH is between 0 and 7% [20,43,46,59,62,63,64,65,66]. Some countries, such as Italy, have guidelines stating that all anticoagulated patients who have undergone a head CT should be observed for 24 h and have a repeat CT before discharge; others suggest discharge only after observation or observation in the community. This is particularly true for older patients for whom an extended stay in an ED can be tiring and can expose them to various risks, from the increased risk of delirium to the risk of death. Despite specialist emergency care, mortality is reportedly high, with up to 34% dying in hospitals [49,67,68,69]. The problem of LOS is well recorded in the literature, and some authors have proposed that patients on anticoagulant therapy may be discharged if the admission CT does not reveal an ICH, provided that they are accompanied by a caregiver and are informed about red flags [20,43,46,59,62,63,64,65,66].

A study showed that the hemorrhages on the control CT scan were only seen under warfarin and this further suggests that this drug might be more frequently associated with delayed bleeding than DOACs [46]. However, we agree with the authors that studies evaluating the impact of DOACs in these patients are practically non-existent, and further research is needed in this area. We believe our analysis makes a contribution.

### 4.3. Comparison of CT at Diagnosis and 24 h Later

The review of our data series shows that the diagnosis of ICH is made in the majority of cases (95.45%) at first CT within the first 6 h of observation. Nevertheless, in a not negligible percentage of patients (4.55%), the diagnosis is made 24 h later with a second CT scan.

In particular, about 1.28% of patients on DOAC and 3.2% of patients on VKAs tested positive at T24. The diagnoses made during the follow-up demonstrate, in our opinion, the importance of observation. It is essential to provide each MHI patient with an information sheet on what to do at home and to ensure that there is a caregiver present [41,51,70]. In adults, observation has been shown to be as effective as the CT in the presence of a caregiver and a patient able to understand home instructions [46]. Our results show similar percentages of positive second CTs of the brain as previous studies [20,43,46,59,62,63,64,65,66]. In addition, even our patients who tested positive for the second CTs did not require surgery, as another study showed. In our observation, the rate of delayed hemorrhage was relatively low. We, therefore, with some prudence, align with other authors that suggest that patients presenting with a negative first CT scan and without neurological deterioration can be safely discharged after a short period of in-ward observation with a low rate of complications and without a second CT scan [20,43,46,59,62,63,64,65,66]. We also think that this could apply especially to patients on DOAC.

### 4.4. Need of Surgery

In our study population, there were very few patients who needed surgery, and there was no statistically significant difference between the therapy groups in the study. No patients in the DOAC group required surgery. Although this correlated with other studies that have seen better outcomes for patients on DOAC than those on VKAs, other studies have demonstrated opposing results [35,36,37]. Larger studies are needed to shed light on this. No statistically significant difference between the groups in anticoagulant therapy emerged regarding Marshall scale and hemorrhage volume. Again, however, we must note that there were only four patients on DOACs and larger cohorts and more studies are needed.

We are the only hospital in the province with a neurosurgery service, and we are one of the six trauma centers in our region (Lombardia). This makes the results of the revisit analysis slightly more reliable. ED revisits were higher in the two groups of patients on anticoagulant therapy compared to the control group. In our opinion, this could be influenced mainly by the age of the various groups. Patients on anticoagulant therapy are older [60], and it has recently been observed that older patients revisit EDs more frequently because of their frailty [49,67,68,69,71]. In our study, none of the patients with an ED revisit within 30 days developed ICH. 

With regard to the need for an extension of the observation in our data, it is clear that this is greater in the DOACs and VkAs groups. However, there are many reasons for the extension of the observation. Please note that elderly patients often depend on the presence of a care-giver or the availability of ambulances for re-entry. The overcrowding of the E.D. also affects the timing [72,73,74]. For this reason, research has also focused on possible solutions to this problem [75,76,77,78].

### 4.5. Intrahospital Mortality

Our study did not see any intrahospital mortality related to MHI. This is in line with the known low short-term mortality of mild head trauma, and it would require a larger cohort to observe any deaths. Even though survival and life expectancy are significantly decreased for patients with MHI compared with non-injured populations [55,56,57,79,80,81,82], MHI confers an increased risk of mortality in the months and years after hospital discharge. Recent studies have increasingly focused on the lasting effects of MHI and observed that chronic diseases were associated with post-discharge mortality after MHI. It has been argued that its effect on older patients could be partly explained by pre-existing comorbidities [50,52,55,82,83,84,85,86,87,88,89,90]. It may be interesting for future studies to assess the possible impact of anticoagulant drugs on long-term mortality, and also the effects that DOACs may have on the overall outcome of emergency and elective neurosurgical pathologies, as raised by our group [48,53,58,91,92,93,94,95]. 

Last but not least in our opinion the value of this study consists in assessment of the role of anticoagulant therapy on the risk of intracranial haemorrhage after mild head trauma in daily clinical practice of the real world.

## 5. Limitations of the Study

Our study has several limitations. First, our conclusions are limited because of the observational nature of the study, including retrospective retrieval of information. For example, we do not know for sure when anticoagulation was restarted in the patients after they were discharged. Second, we did not compare the care patients received. Our outcomes may, therefore, have been affected by the different qualities or timeliness of the treatment. Another limitation of the study is that too few patients needed surgery to be able to give reliable conclusions about this outcome. In addition, the ED revisit data must be interpreted very carefully. The analysis of this data arises from the fact that patients and their caregivers are instructed to return if they experience symptoms related to head trauma. It should not be interpreted as a 30-day follow-up. Patients may, however, have gone independently for multiple reasons (e.g., family or work) outside the province or may have been taken by the territorial service to another center, perhaps because they live closer to it. One limitation of the study that should be noted is that the time from the last dose of a DOAC that a patient took until their arrival to the hospital is not known. Therefore, we do not know how much actual drug/anticoagulation the person had inside their body. This is a limit related to ED environment itself and that should be emphasized: the literature has only recently started considering the importance and relative unreliability of anamnestic data collected from patients presenting at emergency departments (EDs) [96,97].

## 6. Conclusions

In our ED, patients who came for mild head trauma who were on DOAC therapy had a similar prevalence of recurrent or worsened post-traumatic ICH following MHI to patients who were not taking any therapy. Patients who were on VKA therapy had about twice the prevalence of ICH than patients not taking any anti-aggregation or anticoagulant therapy or patients on DOAC. The increased prevalence of ICH also remains when correcting for age. In our cohort of patients, there was no statistically significant difference in the need for a neurosurgical intervention, although an interpretation of this result must consider the very small number of patients needing surgery.

## Figures and Tables

**Table 1 medicina-56-00308-t001:** Clinical criteria for computed tomographies (CTs).

Focal Neurological Signs or Symptoms
Therapy-resistant headache
Drowsiness
Repeated episodes (>2) of vomiting
GCS score < 15
Loss of consciousness
Amnesia before impact
Patients over the age of 65
Major trauma dynamics

HGCS, Glasgow Coma Scale.

**Table 2 medicina-56-00308-t002:** Dynamic criteria for major trauma.

Ejected from the Vehicle
Motorcyclist thrown from the vehicle
Deaths in the same vehicle
Intrusion of the cockpit > 30 cm
Fall from a height > 2 m
Pedestrian projected or rolled or hit at speed > 10 km/h
High-energy impact (speed > 65 km/h)
Vehicle rollover
Extrication time > 20 min

**Table 3 medicina-56-00308-t003:** (a) Main clinical features, (b) Mechanism of injury.

	DOAC (Direct Oral Anticoagulants)	VKAS (Vitamin K Antagonists)	NT (No Therapy)
(**a**) Main clinical features			
number of patients	78	156	1443
HR (mean ± SD) (bpm)	78 ± 18	78 ± 15	80 ± 14
SatO_2_ (mean ± SD) (%)	96 ± 3	96 ± 3	98 ± 2
DBP (mean ± SD) (mmHg)	74 ± 13	73 ± 14	79 ± 12
SBP (mean ± SD) (mmHg)	137 ± 20	134 ± 23	137 ± 22
Age (years)	80 ± 10	82 ± 10	54 ± 23
Age > 65 (%)	94.87	95.51	36.10
Gender (male %)	39	42	51
Major dynamics (%)	1.28	1.28	6.09
GCS < 15 (%)	5.12	3.84	1.66
Patients who underwent CTs (%)	100	100	54.7
(**b**) Mechanism of injury			
Falls (%)	75.00	71.79	42.07
accidental trauma (%)	9.62	5.13	18.85
Violence (%)	0.64	0.00	8.18
minor dynamics road accident (%)	1.28	1.28	7.83
Syncope (%)	8.97	15.38	7.35
major dynamic road accident (%)	0.64	1.28	4.16
seizure in epileptic patient (%)	0	0.00	0.76
sports accident (%)	0	0.00	0.83
other cause (%)	3.85	5.13	9.98

HR, heart rate; DBP, diastolic blood pressure; SBP, systolic blood pressure; SatO_2_, oxygen saturation; SD, standard deviation.

**Table 4 medicina-56-00308-t004:** Regressions by therapy group.

	Odds Ratio	Std. Err.	*z*	*p* > |*z*|	[95% Conf. Interval]	
**NT (1443)**	1 (reference)					
**VKAs (156)**	2.570	0.608	3.99	0.000	1.616	4.088
**DOACs (78)**	0.6638	0.347	−0.78	0.434	0.2376	1.854

**Table 5 medicina-56-00308-t005:** Age-adjusted regressions: association between therapy group and age-adjusted hemorrhage risk.

	Odds Ratio	Std. Err.	*z*	*p* > |*z*|	[95% Conf. Interval]	
**NT (1443)**	1 (reference)					
**VKAs (156)**	1.655	0.422	1.97	0.048	1.003	2.730
**DOACs (78)**	0.4387	0.233	−1.55	0.122	0.154	1.246

**Table 6 medicina-56-00308-t006:** Summary of key outcomes.

	DOAC	VKAs	NT
ICH rate (%)	5.13	17.3	6.51
ICH rate T0 (%)	3.85	14.1	4.44
ICH rate T24 (%)	1.28	3.2	0.07
Need for surgery (%)	0	0.64	0.44
ED revisit within 30 days (%)	14.1	10.9	7.13
Appearance of ICH at ED revisit (%)	0	0	0
Intrahospital mortality (%)	0	0	0

ICH, intracranial hemorrhage; ED, emergency department.

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
