# Peer review of "Rates of Intracranial Hemorrhage in Mild Head Trauma Patients Presenting to Emergency Department and Their Management: A Comparison of Direct Oral Anticoagulant Drugs with Vitamin K Antagonists"

_medicina, 2020, doi:10.3390/medicina56060308_

Round 1
Reviewer 1 Report
Single center, 2 year, retrospective study. N=236 (156 on VKA and 78 on DOAC). Control was 1443.
The primary outcome should be changed to recurrent or worsened post-traumatic ICH following MHI. The original head bleed occurred at the time of the initial injury.
In the inclusion criteria, please define mild head trauma. Also, please define what you mean by “on anticoagulation”. What exact medications were included in the study. Did you include/exclude patients on antiplatelet agents?
Why is an exclusion criteria “clinical suspicion of depressed skull fracture”? Didn’t all of these people undergo CT scan and so one can exclude “presence of depressed skull fracture”? Even so, why exclude this population at all?
One limitation of the study that should be noted is that the time from the last dose of a DOAC that a patient took until their arrival to the hospital is not known. Therefore, we do not know how much actual drug/anticoagulation the person had inside their body.
In table 3, please define “NT”. I assume this is the control group. Also, please note under each heading the number of patients in each group.
Your methods state that patient received vitamin K as the reversal for VKA. This therapy takes a few hours to work. Did they also receive plasma or PCC? If not, why not? When reporting the INR in this group, please include the standard deviation also. Currently, you have only reported the average.
Please include the mechanism of injury for the patients.
In table 4: please include the number of patients in each category
When was anticoagulation restarted in the patients after they were discharged?
How many patients in each group failed the observation therapy and had to be admitted for more observation? You have only reported the need for surgery. What about prolonged observation?
Author Response
We want to thank the Editor-in-Chief and all the Reviewers for their comments and suggestions that have been very precious for us in order to improve the quality and clarity of our manuscript in the context of a challenging topic.
Below, we report an itemized, point-by-point response to the Reviewers’ kind remarks.
All the changes in the text have been reported in track change mode ON.
Reviewer 1
Single center, 2 year, retrospective study. N=236 (156 on VKA and 78 on DOAC). Control was 1443.
- The primary outcome should be changed to recurrent or worsened post-traumatic ICH following MHI. The original head bleed occurred at the time of the initial injury.
kind reviewer, Thank you for the courteous and pertinent remark. Let's correct in the text as it suggests. In particular we would change
- in the introduction: “The primary goal was to determine the difference in the incidence of post-traumatic intracranial bleeding following MHI.”
In: “The primary goal was to determine the difference in the incidence of recurrent or worsened post-traumatic ICH following MHI”
- In study design: “The primary endpoint was the diagnosis of post-traumatic ICH during observation in the Emergency Department.”
In: “The primary endpoint was the diagnosis off recurrent or worsened post-traumatic ICH during observation in the Emergency Department”
- In conclusion: “In our ED, patients who came for mild head trauma who were on DOAC therapy had a similar prevalence of ICH to patients who were not taking any therapy “
In: “In our ED, patients who came for mild head trauma who were on DOAC therapy had a similar prevalence of recurrent or worsened post-traumatic ICH following MHI to patients who were not taking any therapy”
- In the inclusion criteria, please define mild head trauma. Also, please define what you mean by “on anticoagulation”. What exact medications were included in the study. Did you include/exclude patients on antiplatelet agents?
thank you for the comment.
- Regarding the definition of M.H.I. we added in the manuscript:
line 47: “Minor head injury is defined as a patient with a history of loss of consciousness, amnesia, or disorientation and a Glasgow Coma Scale (GCS) score of 13–15.”
Here are the references for clarity:
- I G Stiell, G A Wells, K Vandemheen, C Clement, H Lesiuk, A Laupacis, R D McKnight, R Verbeek, R Brison, D Cass, M E Eisenhauer, G Greenberg, J Worthington. The Canadian CT Head Rule for Patients With Minor Head Injury. Lancet. 2001 May 5;357(9266):1391-6. doi: 10.1016/s0140-6736(00)04561-x. PMID: 11356436 DOI: 10.1016/s0140-6736(00)04561-x”
- S R Shackford 1 , S L Wald, S E Ross, T H Cogbill, D B Hoyt, J A Morris, P A Mucha, H L Pachter, H J Sugerman, K O'Malley, et al. The Clinical Utility of Computed Tomographic Scanning and Neurologic Examination in the Management of Patients With Minor Head Injuries J Trauma. 1992 Sep;33(3):385-94. doi: 10.1097/00005373-199209000-00009. PMID: 1404507 DOI: 10.1097/00005373-199209000-00009”
- regarding the definition of anticoagulant therapy we thought to include in the text:
- line 84: “We excluded patients on antiplatelet or heparin therapy and who did have clotting disorders from both the VkAs and DOACs group.” and
- “for the DOACs group we recruited patients in therapy with apixaban, dabigatran, edoxaban and rivaroxaban. regarding the VkAs group we recruited patients in therapy with Acenocumarol and warfarin”
- Why is an exclusion criteria “clinical suspicion of depressed skull fracture”? Didn’t all of these people undergo CT scan and so one can exclude “presence of depressed skull fracture”? Even so, why exclude this population at all?
Kind reviewer thank you very much for the comment. My sentence was in danger of being misleading. All patients with the clinical suspicion of basic fracture or head vault are undergoing CTs. It did not fall under the criteria of exclusion per self. However, in our population patients who had clinical suspicion already at the medical examination had moderate or severe head trauma; were therefore not included. Among patients with mild head trauma who had a skull fracture, this was diagnosed with CTs and had not been suspected before. Because the sentence is confusing, we thought we'd remove it.
- One limitation of the study that should be noted is that the time from the last dose of a DOAC that a patient took until their arrival to the hospital is not known. Therefore, we do not know how much actual drug/anticoagulation the person had inside their body.
kind reviewer, we thank you for this note. That's true. In addition, in the environment of an ED it is often difficult to know exactly how well the therapy is taken. Literature has only recently started considering the importance and relative unreliability of anamnestic data collected from patients presenting at emergency departments (EDs) [1, 2, 3, 4, 5, 6, 7]. We have therefore decided to add the following paragraph in the limitations of the study:
“"One limitation of the study that should be noted is that the time from the last dose of a DOAC that a patient took until their arrival to the hospital is not known. Therefore, we do not know how much actual drug/anticoagulation the person had inside their body. This is a limit related to ED enviroment itself and that should be emphasized: Literature has only recently started considering the importance and relative unreliability of anamnestic data collected from patients presenting at emergency departments (EDs)."
here we bring you the bibliography for your convenience:
“Lorsbach M, Gillessen A, Revering K, Juhra C. Information on medical history in the emergency department : Influence on therapy and diagnostic decisions. Med Klin Intensivmed Notfmed. 2020 Feb 10. doi: 10.1007/s00063-020-00661-8. Online ahead of print.PMID: 32040681
Gindi M, Oravitz P, Sexton R, et al. Unreliability of reported tetanus vaccination histories. Am J Emerg Med 2005;23:120–122
Brammen D, Dewenter H, Thiemann V, Majeed RW, Xu T, Heitmann KU, Walcher F, Thun S, Röhrig R. Disseminating a Standard for Medical Records in Emergency Departments Among Different Software Vendors Using HL7 CDA. Stud Health Technol Inform. 2017;243:132-136. PMID: 28883186”
- In table 3, please define “NT”. I assume this is the control group. Also, please note under each heading the number of patients in each group.
thanks to the suggestion, we corrected the table as said. (see manuscript pr whole table). here are the modified lines
|
DOAC (direct oral anticoagulants) |
VKAS (vitamin K antagonists) |
NT (no therapy) |
number of patients |
78 |
156 |
1443 |
- Your methods state that patient received vitamin K as the reversal for VKA. This therapy takes a few hours to work. Did they also receive plasma or PCC? If not, why not? When reporting the INR in this group, please include the standard deviation also. Currently, you have only reported the average.
Thank you for pointing it out. In line with yours suggestions, we report the standard deviation at line 147: (±1.15)
- Please include the mechanism of injury for the patients.
Thank you for giving us the opportunity to include this information as well. we would add in the text:
line 139: “mechanism of injury are summarized in table 3 b”
Table 3b. mechanism of injury
|
DOAC (direct oral anticoagulants) |
VKAS (vitamin K antagonists) |
NT (no therapy) |
|
(78) |
(156) |
(1443) |
Falls (%) |
75,00 |
71,79 |
42,07 |
accidental trauma (%) |
9,62 |
5,13 |
18,85 |
Violence (%) |
0,64 |
0,00 |
8,18 |
minor dynamics road accident (%) |
1,28 |
1,28 |
7,83 |
Syncope (%) |
8,97 |
15,38 |
7,35 |
major dynamic road accident (%) |
0,64 |
1,28 |
4,16 |
seizure in epileptic patient (%) |
0 |
0,00 |
0,76 |
sports accident (%) |
0 |
0,00 |
0,83 |
other cause (%) |
3.85 |
5,13 |
9,98 |
- In table 4: please include the number of patients in each category
thanks to the suggestion, we corrected the table as said:
|
Odds Ratio |
Std. Err. |
z |
P>|z| |
[95% Conf. Interval] |
|
NT (1443) |
1 (reference) |
|
|
|
|
|
VKAs (156) |
2.570 |
.608 |
3.99 |
0.000 |
1.616 |
4.088 |
DOACs (78) |
.6638 |
.347 |
-0.78 |
0.434 |
.2376 |
1.854 |
we did the same for table 5:
|
Odds Ratio |
Std. Err. |
z |
P>|z| |
[95% Conf. Interval] |
|
NT (1443) |
1 (reference) |
|
|
|
|
|
VKAs (156) |
1.655 |
.422 |
1.97 |
0.048 |
1.003 |
2.730 |
DOACs (78) |
.4387 |
.233 |
-1.55 |
0.122 |
.154 |
1.246 |
- When was anticoagulation restarted in the patients after they were discharged?
Patients who resigned were re-trusted to their treating physician. The advice in principle of ED doctors was the prompt resumption of anticoagulant therapy in case of course ICH had not occurred. However, since there is no follow-up we are not given to know how things really went. Their treating doctor may have suspended the oral anticoagulant trp due to the risk of falling or for other clinical reasons. Unfortunately, this is within the limits of retrospective study. We then propose to add to line 287 after “…retrival of information.”:
"For example, we don't know for sure when anticoagulation was restarted in the patients after they were discharged"
- How many patients in each group failed the observation therapy and had to be admitted for more observation? You have only reported the need for surgery. What about prolonged observation?
kind reviewer, thank you for the suggestion. In this regard we would like:
- to add in the results, to line 184 "prolonged observation and"; so the paragraph: "3.4. ED revisited within 30 days" would now call : "prolonged observation and ED revisit within 30 days".
- on line 185, we would add : "the percentage of patients who exceeded the expected observation period was 15% in the control group (patients without any therapy), 20% in the VkAs group and 28% in the DOACs group."
- in discussion, in line 274: "With regard to the need for an extension of the observation in our data, it is clear that this is greater in the DOACs and VkAs groups. However, there are many reasons for the extension of the observation. Please note that elderly patients often depend on the presence of a care-giver or the availability of ambulances for re-entry. The overcrowding of the E.D. also affects the timing*. For this reason, research has also focused on possible solutions to this problem#.”
we put here for your convenience the bibliographic references for the statements concerning the discussion:
“*Richardson DB (2002) The access-block effect: relationship between delay to reaching an inpatient bed and inpatient length of stay. Med J Aust 177(9):492–495
*Di Somma S, Paladino L, Vaughan L, Lalle I, Magrini L, Magnanti M. Overcrowding in emergency department: an international issue. Intern Emerg Med 2015;10:171-5.
*Forero R, McCarthy S, Hillman K. Access block and emergency department overcrowding. Available from: http://ccforum.com/content/15/2/216. Accessed 1 Feb 2013.
# Bergs J, Vandijck D, Hoogmartens O, et al. Emergency department crowding: time to shift the paradigm from predicting and controlling to analysing and managing. Int Emerg Nurs 2016;24:74-7.
# Savioli G, Ceresa IF, Manzoni F, Ricevuti G, Bressan MA, Oddone E. Role of a Brief Intensive Observation Area with a Dedicated Team of Doctors in the Management of Acute Heart Failure Patients: A Retrospective Observational Study. Medicina (Kaunas). 2020 May 21;56(5):E251. doi: 10.3390/medicina56050251. PMID: 32455837
# Department of Health (2000) The NHS Plan: a plan for investment, a plan for reform. Department of Health, UK
# Jones P, Schimanski K (2010) The four hour target to reduce emergency department ‘waiting time’: a systematic review of clinical outcomes. Emerg Med Australas 22:391–398. doi:10. 1111/j.1742-6723.2010.01330.x”
[u1]we changed the table as suggested by the reviewers
Reviewer 2 Report
Review of: Mild Head Trauma in Anticoagulated Patients Accessing an Emergency Department: Safety of Direct Anticoagulant Drugs Compared to Traditional Vitamin K Antagonist Anticoagulants.
Overall, this is an interesting article on the safety of DOACs in comparison with VKAs after a mild head injury. Real world data are drawn from a retrospective observational study comparing 234 anticoagulated patients with a control group of non-anticoagulated patients over a time-lapse of three years.
Particularly in consideration of recent published data questioning the security profile of DOACs after head injury (Zeeshan M et al. J Trauma Act Care Sur, 2018), the value of this study consists in supporting their role in daily clinical practice contradicting an increased risk of intracranial haemorrhage after mild head trauma.
Here are my comments:
Do the Authors have any data of the difference of ICH based on the molecule of DOACs (apixaban, dabigatran, edoxaban and rivaroxaban) the patients were treated with?
The Authors have data of the level of anticoagulation (INR values for VKAs at entry) as explained in chapter 3.1.2. Do the Authors have any data of the anti-Xa/anti-IIa level of activity for DOACs at admission as well? If this is the case, they could be integrated in chapter 3.1.1. One consequence of the availability of this data would be to try to find a correlation of the level of the anticoagulation with the prevalence of ICH.
Page 1 (line 23): Please add an -s: …direct oral anticoagulants (DOACs)
Page 1 (line 33): Please cancel “direct oral anticoagulants” and replace with DOACs, as this acronym was already spelled out in line 24.
Page 2 (lines 46-47): I would suggest to briefly provide a definition of mild head injury (MHJ) as some readers may not as familiar as ED doctors are. Does it correspond to the definition of Shackford et al. (in J Trauma 1992:33:382-94) and taken up by the Canadian CT Head Rule for patients with minor head injury (Stiell IG et al in Lancet Neurol 2001:357:1391-96), namely: “Minor head injury is defined as a patient with a history of loss of consciousness, amnesia, or disorientation and a Glasgow Coma Scale (GCS) score of 13–15”?
Page 2 (line 85): I would suggest simplifying the sentence as follows: …without any anticoagulants, anti-aggregation therapy or others clotting disorders.
Did the Authors take patients with known platelets disorders into account?
Page 2 (line 87): Did the Authors exclude patients treated with together anti-aggregation therapy and anticoagulants?
Page 3 (line 103): I recommend to omit “in our ED” because it is redundant.
Page 3 (line 105): I suggest to simplify the sentence as follows: “Our ED is a major trauma’s reference center. We are the only hospital in the province with a 24 h neurosurgery service.”
Page 3 (line 112): I suggest to replace “are presented” with “are listed” in Table 2 in order to avoid a repetition with the sentence before.
Page 4 (line 140): Table 3
Would it not be interesting to add a line with the number of patients?
DOACs VKAs NT
Patients (nr) 78 156 1443
Please correct mistake after HR: replace “mmHg” with “bpm”
Please specify “NT” in the text below the table: no treatment?
Page 4 (line 145): invert the place of the acronym in brackets:…also had their international normalized ratio for prothrombine time (INR) values measured.
Page 4 (line 148): I have a major concern here.
I wonder why the authors chose to stratify the patients with an INR range of 1.5-3, knowing that a therapeutic anticoagulation is defined by an INR between 2.0-3.0. Otherwise, they should briefly explain their choice of an INR range of 1.5-3.
I recommend to clearly specify the three groups (…below the therapeutic range (INR <…), while XX (%) had an INR between…, and 26 patients (18%) above 3.
Page 5 (line 155): I suggest to simplify the title with: 3.2.1 Patients on DOACs
Page 5 (line 161): I suggest to simplify the title with: 3.2.2 Patients on VKAs
Page 5 (line 178): I would completely omit the title 3.3.1, as in the text both aspects, patients on DOACs and on VKAs, are discussed. Therefore, the title 3.3 need for surgery is enough.
Page 6 - 8: Discussion
I would improve the structure of the discussion from a logical point of view. The points discussed are quite clear (ICH, patient management, comparison of CT at diagnosis and 24 hours later, need of surgery and intrahospital mortality), however their paragraph requires being better defined. Moreover, please check time verb tense consistency. The Authors chose to write their article using the present tense, so check for coherence.
Page 6 (lines 196-197):
I suggest to completely cancel the first sentence as it is a repetition of the sentence comprised in lines 203 and 204 and instead directly begin, with the next two sentences as a type of introduction (“It is important…” and “Moderate and…”).
I would first comment on the data of the study and then compare it with the literature. Following this logic, the paragraph comprised between lines 199 and 202, should be placed after the last sentence line 209.
Page 6 (line 203): I suggest to replace “with other populations” with “we recorded a significantly higher bleeding rate (17.3%) compared to the other two groups”.
Page 6 (line 216): Please cancel “of course”. Add an -s to DOAC (=DOACs)
Page 7 (line 222): I would be more categorical and modify the sentence as follows: “Based on our data, DOACs have a better safety prolife than VKAs in terms of ICH following MHI.”
Page 7 (lines 223 - 238): This paragraph is too long and reiterates well-known consequences of older people admitted to an ED or hospital. The points treated are out off topic. In particular, lines between 230 and 238 should/could be summarized in one or two sentences.
Page 7 (line 229): modify as follows:”…This is particularly true for older patients for whom an extended stay in ….”
Page 7 (line 238-239): “A study showed”….signs a beginning of a new paragraph and the sentence should therefore be written on a new line. Please modify this sentence as follows: …, and this further suggests that this drug might be more frequently associated with delayed bleeding than DOACs.
Page 7 (line 242): Please cancel the sentence beginning with: It is…..This sentence is not useful.
Page 7 (line 244): Instead of the latter sentence, begin with: Moreover, the review of ….Modify the following sentence as follows: “Nevertheless, in a not negligible percentage of patients (4.55%), the diagnosis is made 24 hours later with a second CT scan”.
Page 7 (line 262): Please omit “Further larger…”. Instead write: Larger studies are needed…
Page 7 (line 265): Please simply: ….only four patients on DOACs and larger cohorts and more studies are needed.
Page 8 (line 270): Please cancel “in the literature” as it is already clear…
Page 8 (line 299):
In the paragraph conclusions, I would suggest to stress the fact that the almost same prevalence of ICH between DOACs patients and control group is very likely biased (i.e.: less CT scan performed in the control group).
Please replace “for an operating room” with“ in the need of a neurosurgical intervention, although…”.
Finally, I would recommend the Authors to have a look at the following recent articles, which could be easily integrated in the text.
- Bouget, J, Balusson, F, Maignan, M, et al. Major bleeding risk associated with oral anticoagulant in real clinical practice. A multicentre 3‐year period population‐based prospective cohort study. Br J Clin Pharmacol. 2020; 1– 11.
- Vinogradova Y, Coupland C, Hill T, Hippisley‐Cox J. Risks and benefits of direct oral anticoagulants versus warfarin in a real world setting: cohort study in primary care. BMJ. 2018; 362:k2505.
- Ntaios G, Papavasileiou V, Makaritsis K, Vemmos K, Michel P, Lip GYH. Real‐world setting comparison of nonvitamin‐k antagonist oral anticoagulants versus vitamin‐k antagonists for stroke prevention in atrial fibrillation: a systematic review and meta‐analysis. Stroke. 2017; 48(9): 2494‐2503
Author Response
We want to thank the Editor-in-Chief and all the Reviewers for their comments and suggestions that have been very precious for us in order to improve the quality and clarity of our manuscript in the context of a challenging topic.
Below, we report an itemized, point-by-point response to the Reviewers’ kind remarks.
All the changes in the text have been reported in track change mode ON.
Reviewer 2:
- Overall, this is an interesting article on the safety of DOACs in comparison with VKAs after a mild head injury. Real world data are drawn from a retrospective observational study comparing 234 anticoagulated patients with a control group of non-anticoagulated patients over a time-lapse of three years.
Particularly in consideration of recent published data questioning the security profile of DOACs after head injury (Zeeshan M et al. J Trauma Act Care Sur, 2018), the value of this study consists in supporting their role in daily clinical practice contradicting an increased risk of intracranial haemorrhage after mild head trauma.
kind reviewer, we thank you for your careful reading and the precious and welcome comment that we think focuses very well on the design and the results of the study. With your permission we would like to integrate the discussion with this aspect that you emphasized by adding at the end of the discussion itself: "last but not least in our opinion the value of this study consists in assessment of the role of anticoagulant therapy on the risk of intracranial haemorrhage after mild head trauma in daily clinical practice of the real world."
Here are my comments:
- Do the Authors have any data of the difference of ICH based on the molecule of DOACs (apixaban, dabigatran, edoxaban and rivaroxaban) the patients were treated with?
Thank you for the observation, we certainly have the data requested: we then propose to add, to line 154: “2 ICH were patients on Apixaban; 1 on rivaroxaban; and 1 on dabigatran”.
- The Authors have data of the level of anticoagulation (INR values for VKAs at entry) as explained in chapter 3.1.2. Do the Authors have any data of the anti-Xa/anti-IIa level of activity for DOACs at admission as well? If this is the case, they could be integrated in chapter 3.1.1. One consequence of the availability of this data would be to try to find a correlation of the level of the anticoagulation with the prevalence of ICH.
we agree that it would be an interesting fact, however it is not a dosage that is routinely performed in urgency. Data is too few, or rather, rare.
- Page 1 (line 23): Please add an -s: …direct oral anticoagulants (DOACs)
thank you, done. See the comment in the manuscript:
I'm sorry about the TYPO:: “anticoagulants”
- Page 1 (line 33): Please cancel “direct oral anticoagulants” and replace with DOACs, as this acronym was already spelled out in line 24.
thank you, done. See the comment in the manuscript:
in line with the reviewers' suggestions, we would change the text: " Patients on VKA developed ICH more frequently than patients on direct oral anticoagulants (DOACs) …” in “Patients on VKA developed ICH more frequently than patients on DOACs.”
- Page 2 (lines 46-47): I would suggest to briefly provide a definition of mild head injury (MHJ) as some readers may not as familiar as ED doctors are. Does it correspond to the definition of Shackford et al. (in J Trauma 1992:33:382-94) and taken up by the Canadian CT Head Rule for patients with minor head injury (Stiell IG et al in Lancet Neurol 2001:357:1391-96), namely: “Minor head injury is defined as a patient with a history of loss of consciousness, amnesia, or disorientation and a Glasgow Coma Scale (GCS) score of 13–15”?
thank you for the comment, it's correct. See the comment in the manuscript:
in line with reviewer suggestions we would like to specify at this point: “Minor head injury is defined as a patient with a history of loss of consciousness, amnesia, or disorientation and a Glasgow Coma Scale (GCS) score of 13–15.”
Here are the references for clarity:
- I G Stiell, G A Wells, K Vandemheen, C Clement, H Lesiuk, A Laupacis, R D McKnight, R Verbeek, R Brison, D Cass, M E Eisenhauer, G Greenberg, J Worthington. The Canadian CT Head Rule for Patients With Minor Head Injury. Lancet. 2001 May 5;357(9266):1391-6. doi: 10.1016/s0140-6736(00)04561-x. PMID: 11356436 DOI: 10.1016/s0140-6736(00)04561-x”
- S R Shackford 1 , S L Wald, S E Ross, T H Cogbill, D B Hoyt, J A Morris, P A Mucha, H L Pachter, H J Sugerman, K O'Malley, et al. The Clinical Utility of Computed Tomographic Scanning and Neurologic Examination in the Management of Patients With Minor Head Injuries J Trauma. 1992 Sep;33(3):385-94. doi: 10.1097/00005373-199209000-00009. PMID: 1404507 DOI: 10.1097/00005373-199209000-00009”
- Page 2 (line 85): I would suggest simplifying the sentence as follows: …without any anticoagulants, anti-aggregation therapy or others clotting disorders.
kind reviewer, we would like to accept simplification. So we'd change the sentence: “We also included a control group consisting of patients with mild head trauma, age > 18, and GCS score of > 13 on arrival at the ED without anticoagulant or anti-aggregation or heparin therapy and who did not have clotting disorders”
in: “We also included a control group consisting of patients with mild head trauma, age > 18, and GCS score of > 13 on arrival at the ED without any anticoagulants, anti-aggregation therapy or others clotting disorders.”
- Did the Authors take patients with known platelets disorders into account?
kind reviewer, thank you for your precise and timely suggestion. Because there was no patient with known platelets disorders we left the sentence “ without others clotting disorders” without specifying further.
- Page 2 (line 87): Did the Authors exclude patients treated with together anti-aggregation therapy and anticoagulants?
yes we've excluded them. so we would have thought to add at line 84: “We excluded patients on antiplatelet or heparin therapy and who did have clotting disorders from both the VkAs and DOACs group.”
- Page 3 (line 103): I recommend to omit “in our ED” because it is redundant.
kind reviewer, thank you for the recommendation. we did.
- Page 3 (line 105): I suggest to simplify the sentence as follows: “Our ED is a major trauma’s reference center. We are the only hospital in the province with a 24 h neurosurgery service.”
kind reviewer, thank you for the suggestion. we did. we propose to write:
“Our ED is a major trauma’s reference center. We are the only hospital in the province with neurosurgery service. Our neurosurgery service is 24 h ”
the proposal stems from the fact that there are not even 12-hour neurosurgery services in the province.
- Page 3 (line 112): I suggest to replace “are presented” with “are listed” in Table 2 in order to avoid a repetition with the sentence before.
kind reviewer, thank you for the recommendation. we did.
- Page 4 (line 140): Table 3
Would it not be interesting to add a line with the number of patients?
- DOACs VKAs NT
Patients (nr) 78 156 1443
kind reviewer, thank you for the recommendation. we did.
Table 3a. Main clinical features |
|||
|
DOAC (direct oral anticoagulants) |
VKAS (vitamin K antagonists) |
NT (no therapy) |
number of patients |
78 |
156 |
1443 |
HR (mean ± SD) (bpm ) |
78 ± 18 |
78 ± 15 |
80 ± 14 |
Sat O2 (mean ± SD) (%) |
96 ±3 |
96 ± 3 |
98 ± 2 |
DBP (mean ± SD) (mmHg) |
74 ± 13 |
73 ± 14 |
79 ± 12 |
SBP (mean ± SD) (mmHg) |
137 ± 20 |
134 ± 23 |
137 ±22 |
Age (years) |
80 ± 10 |
82 ± 10 |
54 ± 23 |
Age >65 (%) |
94,87 |
95,51 |
36,10 |
Gender (male %) |
39 |
42 |
51 |
Major dynamics (%) |
1,28 |
1,28 |
6,09 |
GCS < 15 (%) |
5,12 |
3,84 |
1,66 |
Patients who underwent CTs (%) |
100 |
100 |
54,7 |
HR, heart rate; DBP, diastolic blood pressure; SBP, systolic blood pressure; SatO2, oxygen saturation; SD, standard deviation |
- Please correct mistake after HR: replace “mmHg” with “bpm”
kind reviewer, thank you for the recommendation. we did.
Table 3a. Main clinical features |
|||
|
DOAC (direct oral anticoagulants) |
VKAS (vitamin K antagonists) |
NT (no therapy) |
number of patients |
78 |
156 |
1443 |
HR (mean ± SD) (bpm ) |
78 ± 18 |
78 ± 15 |
80 ± 14 |
Sat O2 (mean ± SD) (%) |
96 ±3 |
96 ± 3 |
98 ± 2 |
DBP (mean ± SD) (mmHg) |
74 ± 13 |
73 ± 14 |
79 ± 12 |
SBP (mean ± SD) (mmHg) |
137 ± 20 |
134 ± 23 |
137 ±22 |
Age (years) |
80 ± 10 |
82 ± 10 |
54 ± 23 |
Age >65 (%) |
94,87 |
95,51 |
36,10 |
Gender (male %) |
39 |
42 |
51 |
Major dynamics (%) |
1,28 |
1,28 |
6,09 |
GCS < 15 (%) |
5,12 |
3,84 |
1,66 |
Patients who underwent CTs (%) |
100 |
100 |
54,7 |
HR, heart rate; DBP, diastolic blood pressure; SBP, systolic blood pressure; SatO2, oxygen saturation; SD, standard deviation |
- Please specify “NT” in the text below the table: no treatment?
kind reviewer, thank you for the recommendation. we did.
Table 3a. Main clinical features |
|||
|
DOAC (direct oral anticoagulants) |
VKAS (vitamin K antagonists) |
NT (no therapy) |
number of patients |
78 |
156 |
1443 |
HR (mean ± SD) (bpm ) |
78 ± 18 |
78 ± 15 |
80 ± 14 |
Sat O2 (mean ± SD) (%) |
96 ±3 |
96 ± 3 |
98 ± 2 |
DBP (mean ± SD) (mmHg) |
74 ± 13 |
73 ± 14 |
79 ± 12 |
SBP (mean ± SD) (mmHg) |
137 ± 20 |
134 ± 23 |
137 ±22 |
Age (years) |
80 ± 10 |
82 ± 10 |
54 ± 23 |
Age >65 (%) |
94,87 |
95,51 |
36,10 |
Gender (male %) |
39 |
42 |
51 |
Major dynamics (%) |
1,28 |
1,28 |
6,09 |
GCS < 15 (%) |
5,12 |
3,84 |
1,66 |
Patients who underwent CTs (%) |
100 |
100 |
54,7 |
HR, heart rate; DBP, diastolic blood pressure; SBP, systolic blood pressure; SatO2, oxygen saturation; SD, standard deviation |
- Page 4 (line 145): invert the place of the acronym in brackets:…also had their international normalized ratio for prothrombine time (INR) values measured.
kind reviewer, thank you for the recommendation. we did.
- Page 4 (line 148): I have a major concern here.
I wonder why the authors chose to stratify the patients with an INR range of 1.5-3, knowing that a therapeutic anticoagulation is defined by an INR between 2.0-3.0. Otherwise, they should briefly explain their choice of an INR range of 1.5-3.
- I recommend to clearly specify the three groups (…below the therapeutic range (INR <…), while XX (%) had an INR between…, and 26 patients (18%) above 3.
kind reviewer, thank you for the comment that allows us to better specify this point. we answer here to his two notes, as in relation to each other.
we therefore propose to change the sentence:: “Only 16 patients (11 %) were below the therapeutic range, while 102 (71 %) had an INR between 1.5 and 3 and 26 patients (18 %) above 3.”
in: " 19 patients (13%) below the therapeutic range (INR <2), while 99 (69%) had an INR between 2-3, and 26 patients (18%) above 3."
- Page 5 (line 155): I suggest to simplify the title with: 3.2.1 Patients on DOACs
kind reviewer, thank you for the recommendation. we did.
- Page 5 (line 161): I suggest to simplify the title with: 3.2.2 Patients on VKAs
kind reviewer, thank you for the recommendation. we did
- Page 5 (line 178): I would completely omit the title 3.3.1, as in the text both aspects, patients on DOACs and on VKAs, are discussed. Therefore, the title 3.3 need for surgery is enough.
kind reviewer, thank you for the recommendation. we did
- Page 6 - 8: Discussion
- I would improve the structure of the discussion from a logical point of view. The points discussed are quite clear (ICH, patient management, comparison of CT at diagnosis and 24 hours later, need of surgery and intrahospital mortality), however their paragraph requires being better defined. Moreover, please check time verb tense consistency. The Authors chose to write their article using the present tense, so check for coherence.
Thank you for the kind suggestion. We have divided the discussion into the sub-figures corresponding to the logical points for which we have divided it.
Regarding verbal timing we have specifically sent to a website specializes in providing scientific and academic editing services. we attach the editing certificate. We are sorry that it was not adequate but we would not be able to have a new one in the time required by the magazine. we will report the problem to the internet service specializes in providing scientific and academic editing services
- Page 6 (lines 196-197):
I suggest to completely cancel the first sentence as it is a repetition of the sentence comprised in lines 203 and 204 and instead directly begin, with the next two sentences as a type of introduction (“It is important…” and “Moderate and…”).
kind reviewer, thank you for the recommendation. we did
- I would first comment on the data of the study and then compare it with the literature. Following this logic, the paragraph comprised between lines 199 and 202, should be placed after the last sentence line 209.
kind reviewer, thank you for the recommendation. we did
- Page 6 (line 203): I suggest to replace “with other populations” with “we recorded a significantly higher bleeding rate (17.3%) compared to the other two groups”.
kind reviewer, thank you for the recommendation. we did
we changed the sentence: “Regarding patients on dicoumarol treatment, we recorded a significantly higher bleeding rate compared with other populations (17.3 %).”
In: “Regarding patients on dicoumarol treatment, we recorded a significantly higher bleeding rate (17.3%) compared to the other two groups.”
- Page 6 (line 216): Please cancel “of course”. Add an -s to DOAC (=DOACs)
kind reviewer, thank you for the recommendation. we did.
Now you can read: "This data must be contextualized. The population on DOACs….”
- Page 7 (line 222): I would be more categorical and modify the sentence as follows: “Based on our data, DOACs have a better safety prolife than VKAs in terms of ICH following MHI.”
thank you for the precious comment that allows us to be more incisive.
We applied the suggested change
- Page 7 (lines 223 - 238): This paragraph is too long and reiterates well-known consequences of older people admitted to an ED or hospital. The points treated are out off topic. In particular, lines between 230 and 238 should/could be summarized in one or two sentences.
thank you for the suggestion that we have decided to simplify the period in: “This is particularly true for older patients for whom an extended stay in an ED can be tiring and can expose them to various risks, from the increased risk of delirium to the risk of death”
- Page 7 (line 229): modify as follows:”…This is particularly true for older patients for whom an extended stay in ….”
thank you for the suggestion that we have decided to simplify the period in: “This is particularly true for older patients for whom an extended stay in an ED can be tiring and can expose them to various risks, from the increased risk of delirium to the risk of death”
- Page 7 (line 238-239): “A study showed”….signs a beginning of a new paragraph and the sentence should therefore be written on a new line. Please modify this sentence as follows: …, and this further suggests that this drug might be more frequently associated with delayed bleeding than DOACs.
thank you for the suggestion now you can read:
“A study showed that the hemorrhages on the control CT scan were only seen under warfarin and this further suggests that this drug might be more frequently associated with delayed bleeding than DOACs”
- Page 7 (line 242): Please cancel the sentence beginning with: It is…..This sentence is not useful.
kind reviewer, thank you for the suggestion. we did.
- Page 7 (line 244): Instead of the latter sentence, begin with: Moreover, the review of ….Modify the following sentence as follows: “Nevertheless, in a not negligible percentage of patients (4.55%), the diagnosis is made 24 hours later with a second CT scan”.
kind reviewer, thank you for the suggestion. we did.
- Page 7 (line 262): Please omit “Further larger…”. Instead write: Larger studies are needed…
kind reviewer, thank you for the suggestion. we did.
- Page 7 (line 265): Please simply: ….only four patients on DOACs and larger cohorts and more studies are needed.
kind reviewer, thank you for the suggestion. we did.
- Page 8 (line 270): Please cancel “in the literature” as it is already clear…
kind reviewer, thank you for the suggestion. we did.
- Page 8 (line 299):
In the paragraph conclusions, I would suggest to stress the fact that the almost same prevalence of ICH between DOACs patients and control group is very likely biased (i.e.: less CT scan performed in the control group).
kind reviewer, thank you for the suggestion. we did.
we would add the following sentence “the fact that the almost same prevalence of ICH between DOACs patients and control group is very likely biased (i.e.: less CT scan performed in the control group , increased awareness of DOACs patients to show up in case of head trauma).”
- Please replace “for an operating room” with“ in the need of a neurosurgical intervention, although…”.
kind reviewer, thank you for the suggestion. we did.
- Finally, I would recommend the Authors to have a look at the following recent articles, which could be easily integrated in the text.
thank you for the comment, they were interesting and useful readings. We had two of them already. we have gladly accepted your suggestion to include them appropriately in the references
- A* Bouget, J, Balusson, F, Maignan, M, et al. Major bleeding risk associated with oral anticoagulant in real clinical practice. A multicentre 3‐year period population‐based prospective cohort study. Br J Clin Pharmacol. 2020; 1– 11.
- B* Vinogradova Y, Coupland C, Hill T, Hippisley‐Cox J. Risks and benefits of direct oral anticoagulants versus warfarin in a real world setting: cohort study in primary care. BMJ. 2018; 362:k2505.
- C* Ntaios G, Papavasileiou V, Makaritsis K, Vemmos K, Michel P, Lip GYH. Real‐world setting comparison of nonvitamin‐k antagonist oral anticoagulants versus vitamin‐k antagonists for stroke prevention in atrial fibrillation: a systematic review and meta‐analysis. Stroke. 2017; 48(9): 2494‐2503
[u1]we changed the table as suggested by the reviewers
[u2]I’m sorry for the TYPO. we corrected as requested by the reviewer.
[u3]we changed the table as suggested by the reviewers
[u4]I’m sorry for the TYPO. we corrected as requested by the reviewer.
[u5]we changed the table as suggested by the reviewers
[u6]I’m sorry for the TYPO. we corrected as requested by the reviewer.

Round 2
Reviewer 2 Report
The paper is now acceptable for publication.
Author Response
The paper is now acceptable for publication.
kind reviewer,
thank you for appreciating our work.
